# BENCHMARKING MACHINE LEARNING ROBUSTNESS IN COVID-19 SPIKE SEQUENCE CLASSIFICATION

## ABSTRACT

The rapid spread of the COVID-19 pandemic has resulted in an unprecedented amount of sequence data of the SARS-CoV-2 viral genome — millions of sequences and counting. This amount of data, while being orders of magnitude beyond the capacity of traditional approaches to understanding the diversity, dynamics and evolution of viruses, is nonetheless a rich resource for machine learning (ML) and deep learning (DL) approaches as alternatives for extracting such important information from these data. It is of hence utmost importance to design a framework for testing and benchmarking the robustness of these ML and DL approaches.

This paper the first (to our knowledge) to explore such a framework. In this paper, we introduce several ways to perturb SARS-CoV-2 spike protein sequences in ways that mimic the error profiles of common sequencing platforms such as Illumina and PacBio. We show from experiments on a wide array of ML approaches from naive Bayes to logistic regression, that DL approaches are more robust (and accurate) to such adverarial attacks to the input sequences, while $k$-mer based feature vector representations are more robust than the baseline one-hot embedding. Our benchmarking framework may developers of futher ML and DL techniques to properly assess their approaches towards understanding the behaviour of the SARS-CoV-2 virus, or towards avoiding possible future pandemics.

## 1 INTRODUCTION

A novel (RNA) coronavirus was identified in January 2020, which began the COVID-19 pandemic that is still ongoing today. With the help of sequencing technology and phylogenetic analysis, the scientific community disclosed that this novel coronavirus has 50% similarity with the Middle-Eastern Repiratory Syndrome Coronavirus (MERS-CoV), 79% sequencing similarity to Severe Acute Respiratory Syndrome Coronavirus (SARS-CoV) — also known simply as "SARS" — and more than 85% similarity with coronavirusus found in bats. Further studies confirmed that bats are the likely reservoir of these coronaviruses; however, the ecological separation of bats from humans indicates that some other organisms may have acted as intermediate hosts. Considering all scientific evidence, the International Committee on Taxonomy of Viruses named the novel RNA virus SARS-CoV-2 (Wu et al., 2020; Park, 2020; Zhang & Holmes, 2020).

RNA viruses generally introduce errors during replication, the resulting mutations incorporated into the viral genome after repeated replication within a single host, generating a heterogenous population of viral quasispecies. However, SARS-CoV-2 has an excellent proofreading mechanism that encodes a nonstructural protein 14 (nsp14) allowing it to have a 10-fold lower mutation rate than the typical RNA viruses. Epidemiologists estimate that SARS-CoV-2 has 33 genomic mutations per year on average. Some of these mutations are advantageous, leading to the more infections variants of SARS-CoV-2 that continue to emerge (Nelson, 2021). Because of this relatively slow process of accumlating mutations, and the fact that each major variant can be characterized or differentiated by a handfull of mutations (SARS-CoV-2 Variant Classifications and Definitions, 2021), small perturbations (errors) in the sequence should not lead to mistaking one variant from another. Moreover, most of these changes occur in the S gene — the segment of the genome which encodes the surface, or spike, protein (see Figure 1) — hence, characterizing variants using (the resulting tran-

scribed) spike proteins is sufficient for the classification task (SARS-CoV-2 Variant Classifications and Definitions, 2021; Kuzmin et al., 2020).

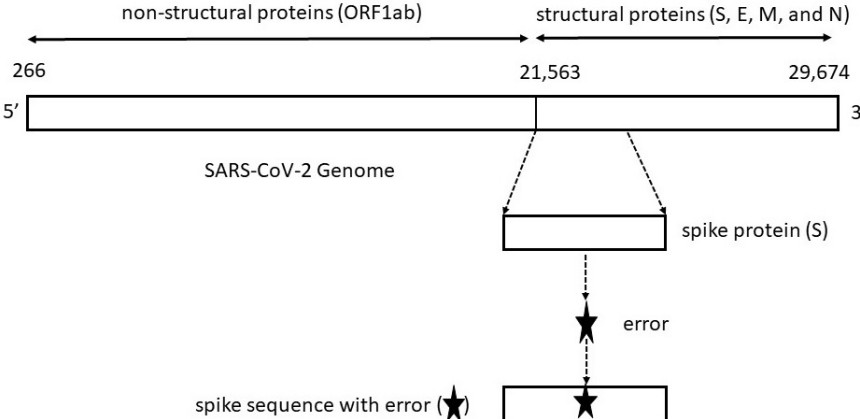

Figure 1: The SARS-CoV-2 genome codes for several proteins, including the surface, or spike protein. Sequence of spike protein is important to understand various molecular mechanisms of SARS-Cov-2. While incorporation of error in spike sequences can bias the final result (Huang et al., 2020).

The diminishing cost of next-generation sequencing (NGS) technology has aided scientists from different parts of the world to generate SARS-CoV-2 whole-genome sequencing (WGS). The Centers for Disease Control and Prevention (CDC) of the United States has also provided a wealth of information on resources, tools, and protocols for SARS-CoV-2 WGS from different sequencing platforms such as Illumina, PacBio, and Ion Torrent. Finally, the Global Initiative on Sharing All Influenza Data (GISAID) hosts the largest SARS-CoV-2 genome sequencing dataset to date — the largest of any virus in history, with millions of sequences. This unprecedented amount of genomic data generation and easy availability allowed researchers to explore the molecular mechanism, genetic variability, evolutionary progress, and capability of development and spread of novel variants of the virus. On the other hand, this amount of data exceeds the capacity of methods such as Nextstrain (Hadfield et al., 2018) or even the more recent IQTREE2 (Minh et al., 2020) by several orders of magnitude — a Big Data challenge. As a result, recent alternative approaches based on clustering and classification of sequences, *e.g.*, to identify major variants, have appeared in the literature (Melnyk et al., 2020; Ali et al., 2021a;b; Ali & Patterson, 2021), with promising accuracy and scalability properties.

Many issues still remain, however, such as sequencing errors being mistaken for mutations in different analyses, when studying the evolutionary and transmission pattern of the SARS-CoV-2 (GISAID History, 2021; Arons et al., 2020), or other viruses. Incorporation of error in NGS sequences due to contamination in sample preparation, sequencing technology, or genome assembly methodology are other confounding factors. Generally, computational biologists filter those sequences having errors or mask those sequence fragments having errors. For example, each GISAID (GISAID Website, 2021) sequence is a consensus sequence from the intra-host viral population sampled from the patient, averaging out the minor variations which exist in this population. While such a consensus sequence is a good representative of this population, *i.e.*, it is still precise enough to capture the SARS-CoV-2 variant harboured by the infected individual, it comes at the cost of losing this important information, such as these minor variations. Such minor variations, when given enough time to evolve, *e.g.*, within an immunocompromised individual, can become dominant — one of the theories theory behind the emergence of the Alpha variant (Frampton et al., 2021).

What this means is that many of machine learning approaches towards clustering and classification of sequences (Ali et al., 2021a;b; Ali & Patterson, 2021) have been operating under rather idealized conditions of virtually error-free consensus sequences. Moveover, these methods rely on a $k$-mer based feature vector representation — an approach that does not even rely on alignment of the sequences, something which can also introduce bias (Golubchik et al., 2007). Such a framework should easily cope with errors as well — something machine learning approaches can do very

naturally (Du et al., 2021). There is hence a great need for some way to reliably benchmark such methods for robustness to errors, which is what we carry out in this paper.

We highlight the main contributions of this paper as follows:

- We propose several ways of introducing errors into spike sequences which reflect the error profiles of modern NGS technologies such as Illumina and PacBio;
- We demonstrate that the $k$-mer based approach is more robust to such errors when compared to the baseline (one-hot encoding); and
- We show that deep learning is generally more robust in handling these errors than machine learning models.

Moreover, we extend our error testing procedure as a framework for bechmarking the performance of different ML methods in terms of classification accuracy and robustness to different types of simulated random errors in the sequences. The two types of errors that we introduce are "consecutive" and "random" errors (see Sec. 3.4). Random errors are just point mutations, which happen uniformly at random along the protein sequence, simulating closely the behaviour of Illumina sequenceing technolgies (Stoler & Nekrutenko, 2021). Consecutive errors, on the other hand, are small subsequences of consective errors, which can model insertion-deletion (indel) errors which are common in third generation long-reads technologies such as Pacific Biosciences (PacBio) SMRT sequencing (Dohm et al., 2020).

This paper is structured as follows. In Sec. 2 we discuss related work. In Sec. 3 we discuss some approaches we benchmark, and then how we benchmark: the type of adversarial attacks we use. Sce. 4 details the experiments, and Sec. 5 gives the results. Finally, we conclude this paper in Sec. 6.

## 2 RELATED WORK

Assessing and benchmarking the robustness of ML or DL approaches by a series of adversarial attacks is popular in the image classification domain (Hendrycks & Dietterich, 2019), but there are others that are closer to the domain of molecular data. In (Schwalbe-Koda et al., 2021), the authors provide a series of realistic adversarial attacks to benchmark methods which predict chemical properties from atomistic simulations *e.g.*, molecular conformation, reactions and phase transitions. Even closer to the subject of our paper — protein sequences – the authors of (Jha et al., 2021) show that methods, such as AlphaFold (Jumper et al., 2021) and RoseTTAFold (Baek et al., 2021) which employ deep neural networks to predict protein conformation are not robust: producing drastically different protein structures as a result of very small biologically meaningful perturbations in the protein sequence. Our approach is similar, albeit with the different goal of classification: namely, to explore how a small number of point mutations (simulating the error introduced certain types of NGS technologies) can affect the downstream classification of different machine learning and deep learning approaches. After getting the numerical representation, a popular approach is to get the kernel matrix and give that matrix as input to traditional machine learning classifiers like support vector machines (SVM) (Leslie et al., 2003; Farhan et al., 2017; Kuksa et al., 2012). However, these methods are expensive in terms of space complexity. Authors in (Ali & Patterson, 2021; Kuzmin et al., 2020) proposes an efficient embedding method for classification and clustering of spike sequences. However, their approaches are either not scalable or perform poorly on bigger datasets.

## 3 PROPOSED APPROACH

In this section, we start by explaining the baseline model for spike sequence classification. After that, we will explain our deep learning model in detail.

### 3.1 ONE-HOT ENCODING (OHE) BASED EMBEDDING

Authors in (Kuzmin et al., 2020) propose that classification of viral hosts of the coronavirus can be done by using spike sequences only. For this purpose, a fixed-length one-hot encoding based feature vector is generated for the spike sequences. In spike sequence, we have 21 unique characters (amino acids) that are *"ACDEFGHIKLMNPQRSTVWXY"*. Also, note that the length of each spike

sequence is 1273 with the stopping character '*' at the $1274^{th}$ position. When we design the OHE based numerical vector for the spike sequence, the length of numerical vector will be $21 \times 1273 = 26733$. This high dimensionality could create the problem of "Curse of Dimensionality (CoD)". To solve CoD problem, any dimensionality reduction method can be used such as Principal Component Analysis (Abdi & Williams, 2010). After reducing the dimensions of the feature vectors, classical Machine Learning (ML) algorithms can be applied to classify the spike sequences. One major problem with such OHE based representation is that it does not preserve the order of the amino acids very efficiently (Ali et al., 2021a). If we compute the pair-wise euclidean distance between any two OHE based vectors, the overall distance will not be effected if a random pair of amino acids are swapped for those two feature vectors. Since the order of amino acids is important in case of sequential data, OHE fails to give us efficient results (Ali et al., 2021a). In this paper, we use OHE as baseline embedding method.

### 3.2 K-MERS BASED REPRESENTATION

A popular approach to preserve the ordering of the sequential information is to take the sliding window based substrings (called mers) of length $k$. This $k$-mers based representation is recently proven to be useful in classifying the spike sequences effectively (Ali et al., 2021a) (see Figure 2 for example of k-mers). In this approach, first, the k-mers of length $k$ are computed for each spike

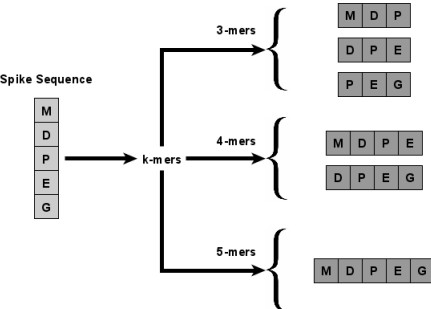

Figure 2: Example of different length $k$-mers in a spike sequence "MDPEG".

sequence. Then a fixed length frequency vector is generated corresponding to each spike sequence, which contains the count of each k-mer in that sequence. One advantage of using k-mers based approach is that it is an *"alignment free"* method unlike OHE, which requires the sequences to be aligned to the reference genome.

**Remark 1** *Sequence alignment is an expensive process and requires reference sequence (genome) (Chowdhury & Garai, 2017; Denti et al., 2021). It may also introduce bias into the result (Golubchik et al., 2007).*

The total number of k-mers in a given spike sequences are:

$$N - k + 1 \tag{1}$$

where $N$ is the length of the sequence. The variable $k$ is the user defined parameter. In this paper, we take $k = 3$ (decided empirically). Since we have 1273 length spike sequences, the total number of k-mers that we can have for any spike sequence is $1273 - 3 + 1 = 1271$.

### 3.2.1 FREQUENCY VECTOR GENERATION

After generating the k-mers, the next step is to generate the fixed-length numerical representation (frequency vector) for the set of k-mers in a spike sequence. Let the set of amino acids in the whole dataset is represented by alphabet $\Sigma$. Now, length of the frequency vector will be $|\Sigma|^k$ (all possible combinations of k-mers in $\Sigma$ of length $k$). Recall that in our dataset, we have 21 unique amino acids in any spike sequence. Therefore, the length of frequency vector in our case would be $21^3 = 9261$ (when we take $k = 3$).

Note that CoD could be a problem in case of k-mers based numerical representation of the spike sequence. To deal with this problem, authors in (Ali & Patterson, 2021) use an approximate kernel method that map such vectors into low dimensional euclidean space using an approach, called Random Fourier Features (RFF) (Rahimi et al., 2007). Unlike kernel trick, which compute the inner product between the lifted data points $\phi$ (i.e. $\langle \phi(a), \phi(b) \rangle = f(a, b)$, where $a, b \in \mathcal{R}^d$ and f(a,b) is any positive definite function), RFF maps the data into low dimensional euclidean inner product space. More formally:

$$z : \mathcal{R}^d \to \mathcal{R}^D \tag{2}$$

RFF tries to approximate the inner product between any pair of transformed points.

$$f(a, b) = \langle \phi(a), \phi(b) \rangle \approx z(a)^T z(b) \tag{3}$$

where $z$ is low dimensional representation. Since $z$ is the approximate representation, we can use it as an input for the classical ML models and analyse their behavior (as done in Spike2Vec (Ali & Patterson, 2021)). However, we show that such approach performs poorly on larger size datasets (hence poor scalability).

### 3.3 KERAS CLASSIFIER

We use a deep learning-based model called the Keras Classification model (also called Keras classifier) to further improve the performance that we got from Spike2Vec. For keras classifier, we use a sequential constructor. It contains a fully connected network with one hidden layer that contains neurons equals to the length of the feature vector (i.e. 9261). We use "rectifier" activation function for this classifier. Moreover, we use "softmax" activation function in the output layer. At last, we use the efficient Adam gradient descent optimization algorithm with "sparse categorical crossentropy" loss function (used for multi-class classification problem). It computes the crossentropy loss between the labels and predictions. The batch size and number of epochs are taken as 100 and 10, respectively for training the model. For the input to this keras classifier, we separately use OHE and $k$-mers based frequency vectors.

**Remark 2** *Note that we are using "sparse categorical crossentropy" rather than simple "categorical crossentropy" because we are using integer labels rather than one-hot representation of labels.*

### 3.4 ADVERSARIAL EXAMPLES CREATION

We use two types of approaches to generate adversarial examples so that we can test the robustness of our proposed model. These approaches are "Random Error generation" and "Consecutive Error generation".

In random error generation, we randomly select a fraction of spike sequences (we call them the set of errored sequences for reference) from the test set (i.e. 5%, 10%, 15%, and 20%). For each of the spike sequence in the set of errored sequences, we randomly select a fraction of amino acids (i.e. 5%, 10%, 15%, and 20%) and flip their value randomly. At the end, we replace these errored sequences set with the corresponding set of original spike sequences in the test set. The ideas is that this simulates the errors made by NGS technologies such as Illumina (Stoler & Nekrutenko, 2021).

In consecutive error generation, the first step is the same as in random error generation (getting random set of spike sequences from the test set "set of errored sequences"). For this set of errored sequences, rather than randomly flipping a specific percentage of amino acid's values for each spike sequence (i.e. 5%, 10%, 15%, and 20%), we flip the values for the same fraction of amino acids but those amino acids are consecutive and at random position in the spike sequence. More formally, it is a consecutive set of amino acids (at random position) in the spike sequence for which we flip the values. At the end, we replace these errored sequences set with the corresponding set of original spike sequences in the test set. The idea is that this simulates indel errors, which are frequently found in third generation long-read technologies such as PacBio (Dohm et al., 2020).

Using the two approaches to generate adversarial examples, we generate a new test set and evaluate the performance of the ML and deep learning models. To measure the performance, we also apply two different strategies. One strategy is called Accuracy and the other is called robustness. In the case of the Accuracy, we compute the average accuracy, precision, recall, F1 (weighted), F1 (Macro), and ROC-AUC for the whole test set including adversarial and non-adversarial examples.

For our second strategy (robustness), we only consider the adversarial examples (set of errored spike sequences) rather than considering the whole test set and compute average accuracy, precision, recall, F1 (weighted), F1 (Macro), and ROC-AUC for them.

## 4 EXPERIMENTAL SETUP

All experiments are conducted using an Intel(R) Xeon(R) CPU E7-4850 v4 @ 2.10GHz having Ubuntu 64 bit OS (16.04.7 LTS Xenial Xerus) with 3023 GB memory. Our pre-processed data is also available online[1], which can be used after agreeing to terms and conditions of GISAID[2]. For the classification algorithms, we use 1% data for training and 99% for testing. Note that our data split and pre-processing follow those of (Ali & Patterson, 2021).

### 4.1 DATASET STATISTICS

We used the (aligned) spike protein from a popular and publicly available database of SARS-CoV-2 sequences, GISAID. In our dataset, we have $2,519,386$ spike sequences along with the COVID-19 variant information. The total number of unique variants in our dataset are $1327$. Since not all variants have significant representation in our data, we only select the variants having more than $10,000$ sequences. After this preprocessing, we are left with $1,995,195$ spike sequences. The dataset statistics for the prepossessed data are given in Table 1. Since most of the variants are new, we do not have all the information available for all them. Therefore, we put "-" in the field in Table 1 for which we do not have any information available online.

Table 1: Dataset Statistics for different variants in our data. The S/Gen. column represents number of mutations on the Spike (S) gene / entire genome. The total number of spike sequences (and corresponding variants) $1,995,195$ after preprocessing (Ali & Patterson, 2021)

| Pango Lineage | Region | Labels | # of Mutations S-gene/Genome | Num. of sequences |
|---|---|---|---|---|
| B.1.1.7 | UK (Galloway et al., 2021) | Alpha | 8/17 | 976077 |
| B.1.351 | South Africa (Galloway et al., 2021) | Beta | 9/21 | 20829 |
| B.1.617.2 | India (Yadav et al., 2021) | Delta | 8/17 | 242820 |
| P.1 | Brazil (Naveca et al., 2021) | Gamma | 10/21 | 56948 |
| B.1.427 | California (Zhang et al., 2021) | Epsilon | 3/5 | 17799 |
| AY.4 | India (SARS-CoV-2 Variant Classifications and Definitions, 2021) | Delta | - | 156038 |
| B.1.2 | - | - | - | 96253 |
| B.1 | | | | 78741 |
| B.1.177 | - | - | - | 72298 |
| B.1.1 | - | | - | 44851 |
| B.1.429 | - | | - | 38117 |
| AY.12 | India (SARS-CoV-2 Variant Classifications and Definitions, 2021) | Delta | - | 28845 |
| B.1.160 | - | | - | 25579 |
| B.1.526 | New York (West Jr et al., 2021) | Iota | 6/16 | 25142 |
| B.1.1.519 | - | - | - | 22509 |
| B.1.1.214 | - | - | - | 17880 |
| B.1.221 | - | - | - | 13121 |
| B.1.258 | - | - | - | 13027 |
| B.1.177.21 | - | - | - | 13019 |
| D.2 | - | - | - | 12758 |
| B.1.243 | - | - | - | 12510 |
| R.1 | - | - | - | 10034 |

---

[1]available in published version
[2]https://www.gisaid.org/

To visualise if there is any (natural) clustering in our data, we generated 2d representation of the k-mers based frequency vectors using t-distributed stochastic neighbor embedding (t-SNE) approach (Van der M. & Hinton, 2008). The t-SNE plot for some of the popular variants is given in Figure 3. Note that we have not given the t-SNE plot for all variants. This is because t-SNE method is computationally very expensive (runtime is $O(N^2)$, where $N$ is the number of data-points (Pezzotti et al., 2016)) and take a lot of time on $\approx 1.9$ million sequences.

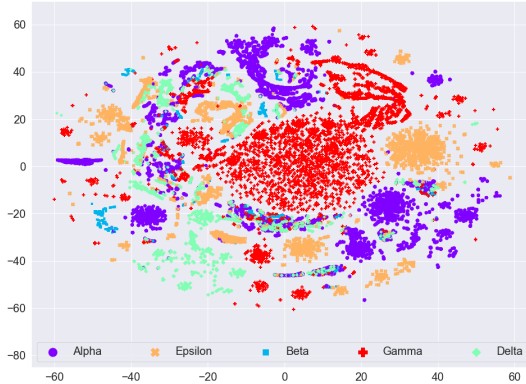

Figure 3: t-SNE plot for the k-mers based feature vectors of spike sequences.

## 5 RESULTS AND DISCUSSION

In this section, we first show the comparison of our deep learning model with the baselines. We then show the results for the two approaches for adversarial examples generation and compare different ML and DL methods. Overall, we elucidate our key findings in the following subsections.

### 5.1 EFFECTIVENESS OF DEEP LEARNING

Table 2 contains the (accuracy) results for our keras classifier and its comparison with different ML models namely Naive Nayes (NB), Logistic Regression (LR), and Ridge Classifier (RC). For keras classifier, we use both OHE and k-mers based embedding approaches separately. We can observe from the results that keras classifier with k-mers based frequency vectors is by far the best approach as compared to the other baselines.

Table 2: Variants Classification Results (1% training set and 99% testing set) for top 22 variants (1995195 spike sequences).

| Approach | Embed. Method | ML Algo. | Acc. | Prec. | Recall | F1 weigh. | F1 Macro | ROC-AUC | Train. runtime (sec.) |
|---|---|---|---|---|---|---|---|---|---|
| ML Algo. | OHE | NB | 0.30 | 0.58 | 0.30 | 0.38 | 0.18 | 0.59 | 1164.5 |
| | | LR | 0.57 | 0.50 | 0.57 | 0.49 | 0.19 | 0.57 | 1907.5 |
| | | RC | 0.56 | 0.48 | 0.56 | 0.48 | 0.17 | 0.56 | 709.2 |
| | Spike2Vec | NB | 0.42 | 0.79 | 0.42 | 0.52 | 0.39 | 0.68 | 1056.0 |
| | | LR | 0.68 | 0.69 | 0.68 | 0.65 | 0.49 | 0.69 | 1429.1 |
| | | RC | 0.67 | 0.68 | 0.67 | 0.63 | 0.44 | 0.67 | 694.2 |
| Deep Learning | One-Hot | Keras Classifier | 0.61 | 0.58 | 0.61 | 0.56 | 0.24 | 0.61 | 28971.5 |
| | k-mers | Keras Classifier | **0.87** | **0.88** | **0.87** | **0.86** | **0.71** | **0.85** | 13296.2 |

To test the robustness of these ML and DL methods, we use both "consecutive error generation" and "random error generation" separately. Table 3 shows the (accuracy) results (using keras classifier with k-mers because that was the best model from Table 2) for the consecutive error generation

method (using different fraction of spike sequences from the test set and different fraction of amino acids flips in each spike sequence). We can observe that keras classifier is able to perform efficiently even with higher proportion of error.

Table 3: Accuracy results for the whole test set (consecutive error seq.) for Keras Classifier with k-mers and different % of errors.

| % of Seq. | % of Error in each Seq. | Acc. | Prec. | Recall | F1 (weighted) | F1 (Macro) | ROC-AUC | Train. runtime (sec.) |
|---|---|---|---|---|---|---|---|---|
| 5 % | 5 % | 0.86 | 0.87 | 0.86 | 0.85 | 0.72 | 0.85 | 15075.72 |
| | 10 % | 0.83 | 0.88 | 0.83 | 0.83 | 0.68 | 0.83 | 15079.3 |
| | 15 % | 0.85 | 0.86 | 0.85 | 0.85 | 0.7 | 0.84 | 13747.5 |
| | 20 % | 0.86 | 0.87 | 0.86 | 0.85 | 0.71 | 0.84 | 11760.21 |
| 10 % | 5 % | 0.85 | 0.87 | 0.85 | 0.84 | 0.68 | 0.83 | 11842 |
| | 10 % | 0.80 | 0.86 | 0.8 | 0.82 | 0.69 | 0.83 | 14658.17 |
| | 15 % | 0.79 | 0.85 | 0.79 | 0.8 | 0.65 | 0.81 | 13783.92 |
| | 20 % | 0.84 | 0.84 | 0.84 | 0.82 | 0.67 | 0.82 | 13159.47 |
| 15 % | 5 % | 0.85 | 0.86 | 0.85 | 0.84 | 0.68 | 0.83 | 15426.38 |
| | 10 % | 0.77 | 0.86 | 0.77 | 0.79 | 0.64 | 0.80 | 8156.5 |
| | 15 % | 0.76 | 0.86 | 0.76 | 0.79 | 0.65 | 0.81 | 16241.72 |
| | 20 % | 0.75 | 0.87 | 0.75 | 0.79 | 0.66 | 0.8 | 15321.63 |
| 20 % | 5 % | 0.73 | 0.86 | 0.73 | 0.77 | 0.65 | 0.80 | 15930.54 |
| | 10 % | 0.76 | 0.87 | 0.76 | 0.79 | 0.64 | 0.79 | 14819.38 |
| | 15 % | 0.76 | 0.88 | 0.76 | 0.79 | 0.65 | 0.80 | 13764.76 |
| | 20 % | 0.80 | 0.81 | 0.80 | 0.77 | 0.63 | 0.78 | 10547.96 |

The robustness results for the consecutive error generation method are given in Table 4. Although we cannot see any clear pattern in this case, the keras classifier is giving us comparatively higher performance in some of the settings.

Table 4: Robustness (only error seq. in test set) results (consecutive error seq.) for Keras Classifier with k-mers and different % of errors.

| % of Seq. | % of Error in each Seq. | Acc. | Prec. | Recall | F1 (weighted) | F1 (Macro) | ROC-AUC | Train. runtime (sec.) |
|---|---|---|---|---|---|---|---|---|
| 5 % | 5 % | 0.60 | 0.57 | 0.60 | 0.54 | 0.10 | 0.55 | 13882.03 |
| | 10 % | 0.15 | 0.62 | 0.15 | 0.1 | 0.04 | 0.52 | 8297.78 |
| | 15 % | 0.12 | 0.51 | 0.12 | 0.03 | 0.01 | 0.5 | 11943.5 |
| | 20 % | 0.60 | 0.44 | 0.60 | 0.48 | 0.07 | 0.53 | 13511.4 |
| 10 % | 5 % | 0.53 | 0.62 | 0.53 | 0.53 | 0.09 | 0.54 | 9475.31 |
| | 10 % | 0.31 | 0.62 | 0.31 | 0.32 | 0.10 | 0.54 | 7137.31 |
| | 15 % | 0.25 | 0.6 | 0.25 | 0.27 | 0.07 | 0.53 | 13399.18 |
| | 20 % | 0.55 | 0.35 | 0.55 | 0.42 | 0.06 | 0.52 | 13232.93 |
| 15 % | 5 % | 0.43 | 0.76 | 0.43 | 0.46 | 0.24 | 0.61 | 13588.11 |
| | 10 % | 0.54 | 0.58 | 0.54 | 0.54 | 0.11 | 0.56 | 14147.05 |
| | 15 % | 0.06 | 0.63 | 0.06 | 0.02 | 0.02 | 0.51 | 13729.87 |
| | 20 % | 0.06 | 0.64 | 0.06 | 0.03 | 0.02 | 0.5 | 13596.25 |
| 20 % | 5 % | 0.17 | 0.68 | 0.17 | 0.18 | 0.10 | 0.54 | 13503.11 |
| | 10 % | 0.48 | 0.63 | 0.48 | 0.48 | 0.09 | 0.55 | 10777.34 |
| | 15 % | 0.47 | 0.59 | 0.47 | 0.49 | 0.09 | 0.55 | 13550.35 |
| | 20 % | 0.49 | 0.39 | 0.49 | 0.33 | 0.03 | 0.50 | 11960.26 |

Table 5 (in appendix) contains the accuracy results for the keras classifier (with k-mers based frequency vectors as input) with random errored sequences approach. We can observe that our DL model is able to maintain higher accuracy even with 20% of the spike sequences having some fraction of error in the test set. Similarly, the robustness results are given in Table 6 (in Appendix).

To visually compare the accuracy of the average accuracy for the consecutive errored sequences approach, we plot the average accuracies in Figure 4a. Similarly, Figure 5 (in Appendix) contains the robustness results for the same two approaches.

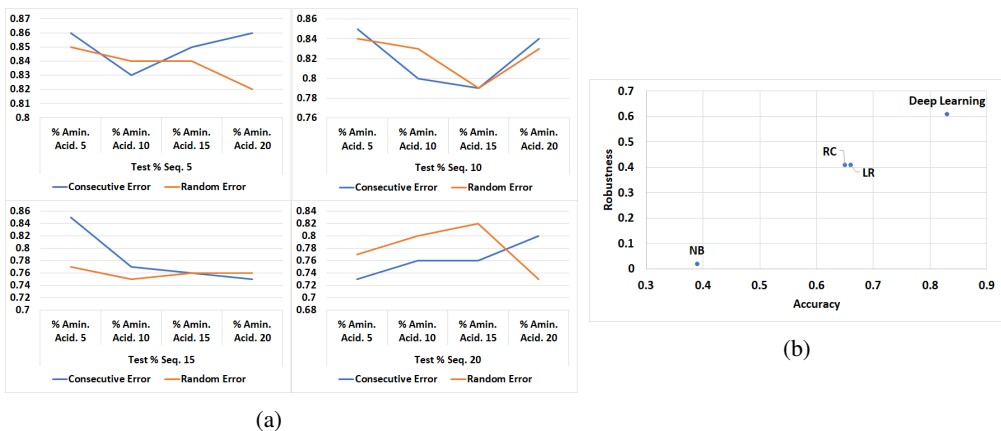

Figure 4: (a) Accuracy Comparison of the Consecutive error generation and Random error generation approaches for different fraction of adversarial spike sequences (note that these results are for one run). (b) Accuracy (x axis) vs Robustness (y axis) plot (for average accuracy values) for different ML and DL methods for 10% adversarial sequences from the test set with 10% amino acids flips.

The accuracy and the robustness results for the other ML models (using Spike2Vec approach) are given in Tables 7 and Table 8, respectively in the appendix. From the results, we can conclude that our deep learning based model is more accurate and robust than other compared machine learning models. Another interesting outcome from the results is that k-mers based feature vector is more robust that traditional one-hot embedding. This is a kind of "proof of concept" that since k-mers preserve the order of the amino acids in a spike sequence (as order matters in genomic sequence data), it outperforms the traditional OHE by a significant margin.

The accuracy vs robustness comparison (for average accuracy values) of different ML and DL methods for 10% adversarial sequences from the test set with 10% amino acids flips is given in Figure 4b. We can see that keras classifier performs best as compared to the other ML methods. This shows that not only our DL method show better predictive performance, but is also more robust as compared to the other ML models.

## 6    CONCLUSION

One interesting future extension is using other alignment free methods such as Minimizers, which have been successful in representing metagenomics data. Since an intra-host viral population can be viewed as a metagenomics sample, this could be appropriate in this context. Another future direction is introducing more adversarial attacks resembling, in more detail, the error profiles of specific sequencing technologies. One could even fine-tune this to the particular experimental setting in which one obtained a sample, similar to sequencing reads simulators such as PBSIM (for PacBio reads).

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

# 7 APPENDIX

Table 5: Accuracy results for the whole test set (random error seq.) for Keras Classifier with k-mers and different % of errors.

| % of Seq. | % of Error in each Seq. | Acc. | Prec. | Recall | F1 (weighted) | F1 (Macro) | ROC-AUC | Train. runtime (sec.) |
|---|---|---|---|---|---|---|---|---|
| | 5 % | 0.85 | 0.88 | 0.85 | 0.84 | 0.69 | 0.84 | 9338.87 |
| 5 % | 10 % | 0.84 | 0.87 | 0.84 | 0.85 | 0.72 | 0.85 | 14365.47 |
| | 15 % | 0.84 | 0.87 | 0.84 | 0.83 | 0.70 | 0.84 | 14996.06 |
| | 20 % | 0.82 | 0.84 | 0.82 | 0.81 | 0.68 | 0.83 | 10958.00 |
| | 5 % | 0.84 | 0.86 | 0.84 | 0.84 | 0.69 | 0.83 | 15465.50 |
| 10 % | 10 % | 0.83 | 0.87 | 0.83 | 0.84 | 0.68 | 0.82 | 15135.49 |
| | 15 % | 0.79 | 0.87 | 0.79 | 0.82 | 0.67 | 0.82 | 14675.58 |
| | 20 % | 0.83 | 0.85 | 0.83 | 0.83 | 0.69 | 0.83 | 14758.57 |
| | 5 % | 0.77 | 0.83 | 0.77 | 0.77 | 0.64 | 0.80 | 16573.58 |
| 15 % | 10 % | 0.75 | 0.83 | 0.75 | 0.77 | 0.66 | 0.80 | 16472.99 |
| | 15 % | 0.76 | 0.86 | 0.76 | 0.79 | 0.65 | 0.80 | 16799.43 |
| | 20 % | 0.76 | 0.84 | 0.76 | 0.77 | 0.67 | 0.81 | 15495.56 |
| | 5 % | 0.77 | 0.87 | 0.77 | 0.81 | 0.67 | 0.81 | 15932.48 |
| 20 % | 10 % | 0.80 | 0.86 | 0.80 | 0.81 | 0.65 | 0.81 | 15823.57 |
| | 15 % | 0.82 | 0.83 | 0.82 | 0.80 | 0.64 | 0.79 | 14597.92 |
| | 20 % | 0.73 | 0.82 | 0.73 | 0.74 | 0.63 | 0.79 | 8885.70 |

Table 6: Robustness results for the whole test set (random error seq.) for Keras Classifier with k-mers and different % of errors.

| % of Seq. | % of Error in each Seq. | Acc. | Prec. | Recall | F1 (weighted) | F1 (Macro) | ROC-AUC | Train. runtime (sec.) |
|---|---|---|---|---|---|---|---|---|
| | 5 % | 0.18 | 0.64 | 0.18 | 0.16 | 0.10 | 0.55 | 11832.46 |
| 5 % | 10 % | 0.08 | 0.65 | 0.08 | 0.06 | 0.02 | 0.51 | 9405.35 |
| | 15 % | 0.28 | 0.58 | 0.28 | 0.28 | 0.07 | 0.53 | 9912.31 |
| | 20 % | 0.56 | 0.36 | 0.56 | 0.43 | 0.06 | 0.52 | 13029.71 |
| | 5 % | 0.62 | 0.59 | 0.62 | 0.54 | 0.10 | 0.55 | 13840.43 |
| 10 % | 10 % | 0.61 | 0.55 | 0.61 | 0.52 | 0.09 | 0.54 | 14016.53 |
| | 15 % | 0.49 | 0.54 | 0.49 | 0.49 | 0.07 | 0.54 | 14038.03 |
| | 20 % | 0.58 | 0.45 | 0.58 | 0.50 | 0.09 | 0.54 | 14202.82 |
| | 5 % | 0.27 | 0.65 | 0.27 | 0.26 | 0.06 | 0.54 | 14790.52 |
| 15 % | 10 % | 0.16 | 0.10 | 0.16 | 0.07 | 0.05 | 0.52 | 14539.64 |
| | 15 % | 0.19 | 0.56 | 0.19 | 0.18 | 0.04 | 0.52 | 13956.71 |
| | 20 % | 0.04 | 0.48 | 0.04 | 0.03 | 0.01 | 0.50 | 13321.69 |
| | 5 % | 0.60 | 0.71 | 0.60 | 0.58 | 0.14 | 0.57 | 14172.11 |
| 20 % | 10 % | 0.22 | 0.58 | 0.22 | 0.19 | 0.08 | 0.53 | 12912.32 |
| | 15 % | 0.46 | 0.57 | 0.46 | 0.43 | 0.05 | 0.52 | 8594.60 |
| | 20 % | 0.03 | 0.59 | 0.03 | 0.01 | 0.01 | 0.50 | 13884.36 |

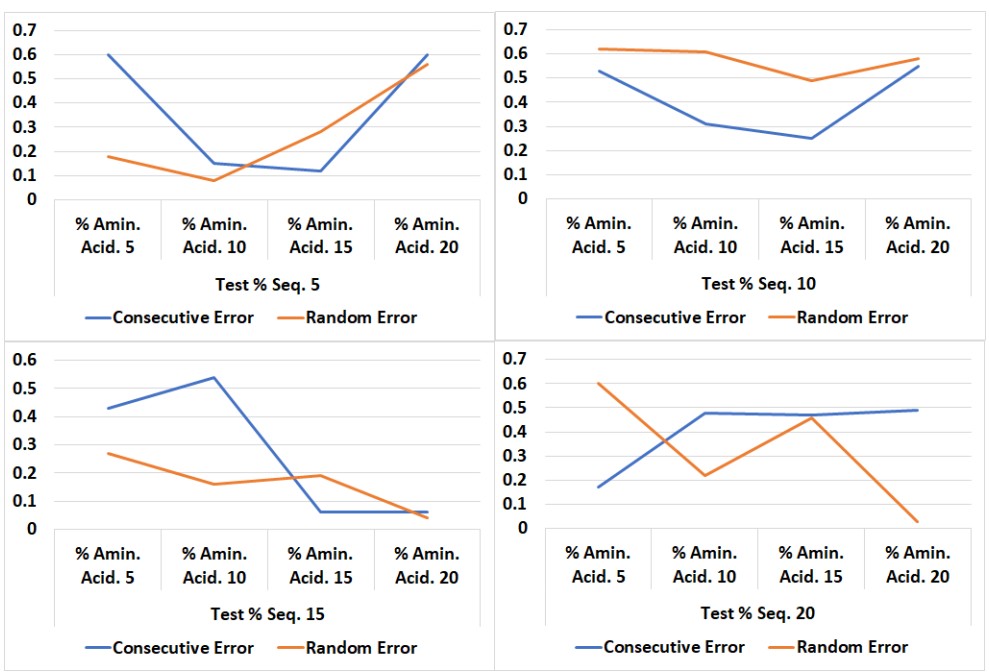

Figure 5: Robustness Comparison of the Consecutive error generation and Random error generation approaches for different fraction of adversarial spike sequences.

Table 7: Accuracy results for the whole test set (random error seq.) for ML models with Spike2Vec approach and different % of errors.

| % of Seq. | % of Error in each Seq. | Acc. | Prec. | Recall | F1 (weighted) | F1 (Macro) | ROC-AUC | Train. runtime (sec.) |
|---|---|---|---|---|---|---|---|---|
| 5 % | 5 % | NB | 0.40 | 0.80 | 0.40 | 0.51 | 0.40 | 0.68 |
| | | LR | 0.68 | 0.68 | 0.68 | 0.64 | 0.68 | 0.69 |
| | | RC | 0.67 | 0.67 | 0.67 | 0.62 | 0.67 | 0.67 |
| | 10 % | NB | 0.40 | 0.80 | 0.40 | 0.51 | 0.40 | 0.68 |
| | | LR | 0.67 | 0.67 | 0.67 | 0.64 | 0.67 | 0.69 |
| | | RC | 0.66 | 0.67 | 0.66 | 0.62 | 0.66 | 0.67 |
| | 15 % | NB | 0.40 | 0.75 | 0.40 | 0.50 | 0.40 | 0.68 |
| | | LR | 0.68 | 0.68 | 0.68 | 0.64 | 0.68 | 0.68 |
| | | RC | 0.67 | 0.67 | 0.67 | 0.62 | 0.67 | 0.67 |
| | 20 % | NB | 0.40 | 0.80 | 0.40 | 0.51 | 0.40 | 0.68 |
| | | LR | 0.67 | 0.67 | 0.67 | 0.64 | 0.67 | 0.69 |
| | | RC | 0.67 | 0.67 | 0.67 | 0.62 | 0.67 | 0.67 |
| 10 % | 5 % | NB | 0.38 | 0.79 | 0.38 | 0.49 | 0.38 | 0.67 |
| | | LR | 0.67 | 0.67 | 0.67 | 0.63 | 0.67 | 0.68 |
| | | RC | 0.66 | 0.66 | 0.66 | 0.61 | 0.66 | 0.66 |
| | 10 % | NB | 0.39 | 0.80 | 0.39 | 0.49 | 0.39 | 0.67 |
| | | LR | 0.66 | 0.66 | 0.66 | 0.62 | 0.66 | 0.68 |
| | | RC | 0.65 | 0.64 | 0.65 | 0.60 | 0.65 | 0.66 |
| | 15 % | NB | 0.38 | 0.80 | 0.38 | 0.49 | 0.38 | 0.67 |
| | | LR | 0.67 | 0.67 | 0.67 | 0.63 | 0.67 | 0.68 |
| | | RC | 0.65 | 0.66 | 0.65 | 0.61 | 0.65 | 0.66 |
| | 20 % | NB | 0.66 | 0.66 | 0.66 | 0.62 | 0.46 | 0.68 |
| | | LR | 0.65 | 0.66 | 0.65 | 0.61 | 0.43 | 0.66 |
| | | RC | 0.36 | 0.79 | 0.36 | 0.46 | 0.36 | 0.66 |
| 15 % | 5 % | NB | 0.36 | 0.79 | 0.36 | 0.46 | 0.36 | 0.66 |
| | | LR | 0.65 | 0.66 | 0.65 | 0.61 | 0.65 | 0.67 |
| | | RC | 0.65 | 0.64 | 0.65 | 0.60 | 0.65 | 0.65 |
| | 10 % | NB | 0.36 | 0.75 | 0.36 | 0.46 | 0.36 | 0.66 |
| | | LR | 0.66 | 0.66 | 0.66 | 0.61 | 0.66 | 0.67 |
| | | RC | 0.65 | 0.65 | 0.65 | 0.60 | 0.65 | 0.65 |
| | 15 % | NB | 0.36 | 0.80 | 0.36 | 0.47 | 0.36 | 0.66 |
| | | LR | 0.64 | 0.65 | 0.64 | 0.60 | 0.45 | 0.67 |
| | | RC | 0.63 | 0.63 | 0.63 | 0.58 | 0.40 | 0.65 |
| | 20 % | NB | 0.36 | 0.75 | 0.36 | 0.46 | 0.35 | 0.66 |
| | | LR | 0.65 | 0.65 | 0.65 | 0.61 | 0.44 | 0.67 |
| | | RC | 0.64 | 0.64 | 0.64 | 0.59 | 0.41 | 0.65 |
| 20 % | 5 % | NB | 0.34 | 0.80 | 0.34 | 0.45 | 0.34 | 0.65 |
| | | LR | 0.63 | 0.64 | 0.63 | 0.59 | 0.63 | 0.66 |
| | | RC | 0.63 | 0.63 | 0.63 | 0.58 | 0.63 | 0.64 |
| | 10 % | NB | 0.34 | 0.75 | 0.34 | 0.45 | 0.34 | 0.65 |
| | | LR | 0.64 | 0.65 | 0.64 | 0.60 | 0.64 | 0.66 |
| | | RC | 0.63 | 0.63 | 0.63 | 0.58 | 0.63 | 0.64 |
| | 15 % | NB | 0.34 | 0.75 | 0.34 | 0.44 | 0.34 | 0.65 |
| | | LR | 0.62 | 0.63 | 0.62 | 0.58 | 0.43 | 0.66 |
| | | RC | 0.60 | 0.60 | 0.60 | 0.56 | 0.39 | 0.64 |
| | 20 % | NB | 0.34 | 0.8 | 0.34 | 0.45 | 0.34 | 0.65 |
| | | LR | 0.64 | 0.64 | 0.64 | 0.6 | 0.43 | 0.66 |
| | | RC | 0.61 | 0.61 | 0.61 | 0.56 | 0.39 | 0.64 |

Table 8: Robustness results for the whole test set (random error seq.) for ML models with Spike2Vec approach and different % of errors.

| % of Seq. | % of Error in each Seq. | Acc. | Prec. | Recall | F1 (weighted) | F1 (Macro) | ROC-AUC | Train. runtime (sec.) |
|---|---|---|---|---|---|---|---|---|
| 5 % | 5 % | NB | 0.02 | 0.03 | 0.02 | 0.01 | 0.02 | 0.50 |
| | | LR | 0.46 | 0.29 | 0.46 | 0.35 | 0.46 | 0.50 |
| | | RC | 0.46 | 0.28 | 0.46 | 0.34 | 0.46 | 0.50 |
| | 10 % | NB | 0.02 | 0.00 | 0.02 | 0.00 | 0.02 | 0.50 |
| | | LR | 0.41 | 0.27 | 0.41 | 0.32 | 0.41 | 0.50 |
| | | RC | 0.43 | 0.28 | 0.43 | 0.33 | 0.43 | 0.50 |
| | 15 % | NB | 0.02 | 0.05 | 0.02 | 0.01 | 0.02 | 0.50 |
| | | LR | 0.46 | 0.28 | 0.46 | 0.34 | 0.46 | 0.50 |
| | | RC | 0.46 | 0.28 | 0.46 | 0.34 | 0.46 | 0.50 |
| | 20 % | NB | 0.02 | 0.03 | 0.02 | 0.00 | 0.02 | 0.50 |
| | | LR | 0.41 | 0.26 | 0.41 | 0.31 | 0.41 | 0.50 |
| | | RC | 0.41 | 0.25 | 0.41 | 0.31 | 0.41 | 0.50 |
| 10 % | 5 % | NB | 0.02 | 0.01 | 0.02 | 0.01 | 0.02 | 0.50 |
| | | LR | 0.46 | 0.29 | 0.46 | 0.35 | 0.46 | 0.50 |
| | | RC | 0.47 | 0.30 | 0.47 | 0.36 | 0.47 | 0.50 |
| | 10 % | NB | 0.02 | 0.00 | 0.02 | 0.00 | 0.02 | 0.50 |
| | | LR | 0.41 | 0.27 | 0.41 | 0.31 | 0.41 | 0.50 |
| | | RC | 0.41 | 0.27 | 0.41 | 0.31 | 0.41 | 0.50 |
| | 15 % | NB | 0.02 | 0.01 | 0.02 | 0.01 | 0.02 | 0.50 |
| | | LR | 0.41 | 0.26 | 0.41 | 0.31 | 0.41 | 0.50 |
| | | RC | 0.42 | 0.26 | 0.42 | 0.31 | 0.42 | 0.50 |
| | 20 % | NB | 0.02 | 0.01 | 0.02 | 0.01 | 0.01 | 0.5 |
| | | LR | 0.47 | 0.3 | 0.47 | 0.36 | 0.04 | 0.51 |
| | | RC | 0.48 | 0.31 | 0.48 | 0.37 | 0.05 | 0.51 |
| 15 % | 5 % | NB | 0.02 | 0.03 | 0.02 | 0.01 | 0.02 | 0.50 |
| | | LR | 0.46 | 0.29 | 0.46 | 0.35 | 0.46 | 0.50 |
| | | RC | 0.41 | 0.26 | 0.41 | 0.31 | 0.41 | 0.50 |
| | 10 % | NB | 0.02 | 0.03 | 0.02 | 0.01 | 0.02 | 0.50 |
| | | LR | 0.41 | 0.27 | 0.41 | 0.32 | 0.41 | 0.50 |
| | | RC | 0.37 | 0.26 | 0.37 | 0.30 | 0.37 | 0.50 |
| | 15 % | NB | 0.02 | 0.01 | 0.02 | 0.01 | 0.01 | 0.50 |
| | | LR | 0.48 | 0.31 | 0.48 | 0.36 | 0.05 | 0.51 |
| | | RC | 0.48 | 0.31 | 0.48 | 0.36 | 0.05 | 0.51 |
| | 20 % | NB | 0.01 | 0.02 | 0.01 | 0.01 | 0.01 | 0.50 |
| | | LR | 0.30 | 0.22 | 0.30 | 0.25 | 0.03 | 0.49 |
| | | RC | 0.36 | 0.24 | 0.36 | 0.29 | 0.03 | 0.50 |
| 20 % | 5 % | NB | 0.02 | 0.03 | 0.02 | 0.01 | 0.02 | 0.50 |
| | | LR | 0.36 | 0.25 | 0.36 | 0.28 | 0.36 | 0.49 |
| | | RC | 0.36 | 0.24 | 0.36 | 0.28 | 0.36 | 0.49 |
| | 10 % | NB | 0.02 | 0.02 | 0.02 | 0.01 | 0.02 | 0.50 |
| | | LR | 0.36 | 0.25 | 0.36 | 0.29 | 0.36 | 0.50 |
| | | RC | 0.36 | 0.26 | 0.36 | 0.28 | 0.36 | 0.49 |
| | 15 % | NB | 0.03 | 0.03 | 0.03 | 0.01 | 0.01 | 0.50 |
| | | LR | 0.41 | 0.28 | 0.41 | 0.32 | 0.04 | 0.50 |
| | | RC | 0.46 | 0.29 | 0.46 | 0.35 | 0.04 | 0.50 |
| | 20 % | NB | 0.02 | 0.06 | 0.02 | 0.01 | 0.01 | 0.50 |
| | | LR | 0.47 | 0.31 | 0.47 | 0.36 | 0.04 | 0.51 |
| | | RC | 0.42 | 0.29 | 0.42 | 0.33 | 0.04 | 0.50 |

