# OpenReview forum: "Benchmarking Machine Learning Robustness in Covid-19 Spike Sequence Classification"
_ICLR.cc/2022/Conference — ICLR 2022 Submitted_

### Official Review · Reviewer_xx9i · 2021-10-28

**Correctness:** 2
**Technical Novelty And Significance:** 2
**Empirical Novelty And Significance:** 2
**Recommendation:** 1
**Confidence:** 5

**Main Review:**

Major comments:

-- The related work lacks previous papers that have tried to tackle the COVID-19 sequence classification task as well as details about what existing methods have been explored for it.

-- One hot encoding representation of sequences has been used as inputs into CNN-based deep learning models. The convolution filters allow the model to extract relevant motif/patterns to make accurate predictions. The deep learning model explored in the paper is just a fully connected network that may not be able to leverage the One hot encoding effectively.

-- The machine learning methods chosen for benchmarking are all simple linear models. K-mer representation has been shown to perform well with SVMs (in the works cited in the paper). Why were these non-linear methods and others like random forests not explored for this study?

-- The k-mer representation presented in this paper is a simple approach and does not account for mismatches or gaps. While such representations have been shown to give better results, they have not been explored in the paper.

-- Each representation has its strengths and weaknesses combined with a relevant model. This aspect needs to be investigated further to support the claims of the paper

-- The paper is missing details on how the hyperparameter tuning was performed for all the models

-- It is unclear why a training/test split of 1% and 99% were chosen for this study. Usually one picks a larger percentage of training samples.

-- The description of the results and the figures are hard to parse.

-- The paper requires thorough proofreading to address the grammatical errors and issues with clarity


**Summary Of The Paper:**

This paper presents a framework to test the accuracy and robustness of different machine learning algorithms in classifying the COVID-19 spike sequences. It benchmarks Naive Bayes, Logistic regression, Ridge regression, and fully-connected neural network architectures. It also explores k-mer and one hot encoding representation of the sequences as inputs into these models. For benchmarking robustness, the paper proposes 2 ways to introduce errors in the spike sequence (reflecting sequencing errors produced by state-of-the-art sequencing technologies).


**Summary Of The Review:**

While the paper aims to investigate a relevant problem, the ICLR venue may not be the best place to present the benchmarking work. It lacks new contributions towards the method or application domains. The paper also requires major revisions and additional results to improve its clarity and to support its claims.

---

> ### Author Response · Authors · 2021-11-21
> **Response to Reviewer xx9i**
>
> (a) Comment: The related work lacks previous papers that have tried to tackle the COVID-19 sequence classification task as well as details about what existing methods have been explored for it.
>
> •Response: It is true that the manuscript does not contain an extensive literature review of recent papers, we however discussed two most recent papers (Ali & Patterson, 2021;Kuzmin et al., 2020) that tackle COVID-19 sequence classification tasks.
>
> (b)Comment: One hot encoding representation of sequences has been used as inputs into CNN-based deep learning models. The convolution filters allow the model to extract relevant motif/patterns to make accurate predictions. The deep learning model explored in the paper is just a fully connected network that may not be able to leverage the One hot encoding effectively.
>
> •Response: It is true that a fully connected network may not be able to leverage OHEeffectively. We will use deeper models in the future to further analyze the performance of OHE.
>
> (c)Comment: The machine learning methods chosen for benchmarking are all simple linear models. K-mer representation has been shown to perform well with SVMs (in the works cited in the paper). Why were these non-linear methods and others like random forests not explored for this study?
>
> •Response: It is true that we used simple linear classifiers in this study.  The other sophisticated methods like SVM could perform better than the used models. However, they were taking more than a day (24 hours) to compute results. Therefore, we only used those classifiers that gave results in a reasonable amount (<24h) of time.
>
> (d)Comment: The k-mer representation presented in this paper is a simple approach and does not account for mismatches or gaps. While such representations have been shown to give better results, they have not been explored in the paper.
>
> •Response: It is true that taking into account mismatches or gaps could improve the results. However, we would like to point out that the main goal of this paper is to show benchmark results to see the performance of traditional classifiers. We will use such sophisticated approaches in the future to further analyze the behavior of classifiers.
>
> (e)Comment: Each representation has its strengths and weaknesses combined with a relevant model. This aspect needs to be investigated further to support the claims of the paper
>
> •Response: We thank the reviewer to point out such an important point. We will include more discussion regarding results in our paper.
>
> (f)Comment: The paper is missing details on how the hyperparameter tuning was performed for all the models
>
> •Response: We thank the reviewer for highlighting this issue. We will include hyperparameter details in the paper.
>
> (g)Comment: It is unclear why a training/test split of 1% and 99% were chosen for this study. Usually one picks a larger percentage of training samples.
>
> •Response: The goal of this study is to see how much information can we extract by using the minimum amount of data for training. That is why we wanted to use 1% data for training to see how classifiers could perform with limited data.
>
> (h)Comment: The description of the results and the figures are hard to parse.
>
> •Response: We thank the reviewer for pointing this out. We will improve the presentation of results.
>
> (i)Comment: The paper requires thorough proofreading to address the grammatical errors and issues with clarity
>
> •Response:We thank the reviewer for pointing this out. We will improve the overall presentation of the paper.

---

> > ### Comment · Reviewer_xx9i · 2021-11-28
> > **Response to the rebuttal**
> >
> > Thank you for providing responses to my comments. Based on all the reviews and the rebuttal, the paper in its current form requires major revisions to be truly useful and impactful for the community. Therefore, I would like to retain my original score.

---

### Official Review · Reviewer_uALA · 2021-10-30

**Correctness:** 1
**Technical Novelty And Significance:** 1
**Empirical Novelty And Significance:** 2
**Recommendation:** 1
**Confidence:** 4

**Main Review:**

Strengths
1. The paper is clear about its contributions and provides a clear introduction to the problem domain.
2. The motivation for the research of using this large dataset to explore and characterise errors in sequencing platforms is reasonable.

Weaknesses:
1. Why is a fixed length sequence of 1273 amino acids considered for the spike protein? Insertions or deletions mean this could be a variable length.
2. The approach does not take into account any domain knowledge from biology. For example, considering the secondary protein structure among the sequences such as the alpha-helices or beta-sheets would potentially be more predictive than a one-hot encoding or k-mer representation. How much structure is conserved among spike proteins? If there is strong conservation this should be used to perform feature selection on the input data.
3. PCA is mentioned but it is unclear whether the input dimension is reduced or to what level.
4. It is unclear how many labels are used in the classification problem. Is it the number of variants (22) or the number of labeled variants (6)? What about the unlabelled variants? Related to this, justification for the topology of the neural network is unclear. Did one hidden layer of 9261 nodes perform better than a deeper model?
5. It would be far superior to statistically model the error profiles from Illumina and PacBio technologies. The proposed approach is simplistic.
6. Why was only 1% of the data set used for training? Such a small proportion of the data in this very imbalanced problem risks not adequately representing all classes.
7. The proportion of labels in table 1 do not match the plot in fig 1. Gamma seems over represented compared to alpha and delta.
8. Use of the performance metrics is unclear and inappropriate for this imbalanced problem. For example, average accuracy will overemphasise classification it the major class at the expense of the smaller classes. Similarly, high weighted F1 values overemphasise the large classes. The macro F1 shows that overall the classification of the minor classes seems to be worse. How was ROC-AUC computed for this multi-class problem?
9. Error bars or confidence should be generated for algorithms so that they can better be compared.

**Summary Of The Paper:**

Taking sequences of the spike portion of COVID-19 genome sequences, this paper builds classifiers for different labeled variants. Two approaches for generating errors in test sequences are introduced and evaluations of the models robustness to these errors measured.

**Summary Of The Review:**

The paper takes a very simplistic approach to classification of these sequences not taking into account any biological domain knowledge. There are concerns about justification for the proposed model, the experimental methodology and evaluation.

---

> ### Author Response · Authors · 2021-11-21
> **Response to Reviewer uALA**
>
> (a) Comment: Why is a fixed-length sequence of 1273 amino acids considered for the spike protein? Insertions or deletions mean this could be a variable length.
>
> •Response: The reviewer has highlighted an important point. The spike sequence dataset that we downloaded from GISAID dataset contains different length sequences. However, the majority (approximately 2.5 million) sequences had length 1273 (plus an ending character *). Since One-Hot Embedding (OHE) works with the fix length vector, we only took the 1273 length spike sequences and discarded the others for this study so that we can compare the k-mers based embedding with the baseline OHE.
>
> (b)Comment: The approach does not take into account any domain knowledge from biology. For example, considering the secondary protein structure among the sequences such as the alpha-helices or beta-sheets would potentially be more predictive than a one-hot encoding or k-mer representation. How much structure is conserved among spike proteins? If there is strong conservation this should be used to perform feature selection on the input data.
>
> •Response: This is true that considering domain knowledge and other efficient feature selection approaches could help to improve the classification performance of underlying ML and DL classifiers. However, we would like to point out that the goal of this study is to benchmark the performance of the classifiers based purely on sequence content (such secondary structure itself is predicted by softwares such as Alphafold and Rosettafold, for example) so that other researchers can compare their findings with these benchmark results.
>
> (c)Comment: PCA is mentioned but it is unclear whether the input dimension is reduced or to what level.
>
> •Response: We did not use PCA to reduce the input dimensions as it is very expensive and take a lot of time on multi-millions of sequences. We rather used a relatively new concept of Random Fourier Feature (RFF) (https://citeseerx.ist.psu.edu/viewdoc/download?doi=10.1.1.145.8736&rep=rep1&type=pdf) to reduce the dimensions of input data. We took 500 dimensions in our study, which is decided empirically. We will make the article clearer so that the reader is not led to think that we used PCA.
>
> (d)Comment: It is unclear how many labels are used in the classification problem. Is it the number of variants (22) or the number of labeled variants (6)? What about the unlabelled variants? Related to this, justification for the topology of the neural network is unclear. Did one hidden layer of 9261 nodes perform better than a deeper model?
>
> •Response: We use 22 variants for the classification (mentioned in table 1). We want to mention that there is no unlabeled data used in this study. In this paper, we just used one deep learning model (Keras classifier). We did not compare it with other deeper models since our goal was to evaluate the performance on non-sophisticated algorithms as part of benchmarking. We will clarify this as well, to reduce confusion in the future.
>
> (e)Comment: It would be far superior to statistically model the error profiles from Illumina and PacBio technologies. The proposed approach is simplistic.
>
> •Response: We agree with the reviewer. In the future, we will use these statistical models for error profiling.
>
> (f)Comment: Why was only 1% of the data set used for training? Such a small proportion of the data in this very imbalanced problem risks not adequately representing all classes.
>
> •Response: Using the minimum amount of data for training (1 %) was enough for us to train the models so that we can get reasonable accuracy (>80 %). After that, we focus on studying the robustness of such high accuracies with this minimum training set, which is the main goal of this paper.
>
> (g)Comment: The proportion of labels in table 1 do not match the plot in fig 1. Gamma seems over represented compared to alpha and delta.
>
> •Response: We believe there is some confusion. Figure 1 is about showing the position of spike sequence in the whole genome. We will try to make this clearer; thank you for pointing this out.

---

> > ### Author Response · Authors · 2021-11-21
> > **Response to Reviewer uALA**
> >
> > (h)Comment: The use of the performance metrics is unclear and inappropriate for this imbalance problem. For example, average accuracy will overemphasize classification it the major class at the expense of the smaller classes. Similarly, high weighted F1 values overemphasize the large classes. The macro F1 shows that overall the classification of the minor classes seems to be worse. How was ROC-AUC computed for this multi-class problem?
> >
> > •Response: We use a combination of metrics to see the performance of classifiers from all aspects. It is true that macro F1 is bad in some cases, but for other scenarios (using Keras classifier for example), it is greater than 70% which shows a reasonable performance of the classifier. For ROC-AUC, we use the one-vs-rest approach for the multi-class problem. We also want to mention here that our main goal is to focus on robustness rather than accuracies. That is the reason why we did not focus much on discussing these evaluation metrics but rather focused more on the performance of models while introducing different fractions of errors in the sequences.
> >
> > (i) Comment: Error bars or confidence should be generated for algorithms so that they can better be compared.
> >
> > •Response: We thank the reviewer for pointing this out. We will include these in the future.

---

> > > ### Comment · Reviewer_uALA · 2021-11-30
> > > **Response to rebuttal**
> > >
> > > Thank you for responding to my comments and questions. However, I still believe there would be major changes required to bring the paper to the appropriate level. So, I maintain my ranking.

---

### Official Review · Reviewer_YVK3 · 2021-11-03

**Correctness:** 2
**Technical Novelty And Significance:** 1
**Empirical Novelty And Significance:** 1
**Recommendation:** 3
**Confidence:** 5

**Main Review:**

While I appreciate the time and energy the authors have put into this paper, I have some major concerns with this paper:

- My major issue with this paper is the sequencing errors are not modeled realistically. Most Illumina sequencers have error rates of less than 0.5% of nucleotides [PMID: 33817639], the authors introduce at least 5% error rates in the amino acids (each amino acid is encoded by three nucleotides). NB. Interestingly, this is the citation that the authors use to justify their “simulated” error rates. The authors mention PacBio sequencing errors, but I see no evidence of that in the text.
- Although spike protein is 1,273 amino acids long; however, there are around 200 amino acids in the receptor-binding domain (RBD) of the spike protein that is of special interest [PMID: 34075212]. A more realistic analysis would focus mostly on the RBD. The community is increasingly generating more high-throughput datasets assaying different aspects of the RBD and ACE2 binding, which can help researchers train more accurate models, e.g., [PMID: 32841599].
- The concepts introduced in the paper are not novel and have been studied in detail: encoding of amino acids, _k_-mer-based models, and simple MLPs have been thoroughly studied in computational genomics.
- The text is hard to follow and has quite a few grammatical errors/missing words starting from the abstract. A thorough review and polish of the text would improve the paper.


**Summary Of The Paper:**

In this paper, the authors propose a method to identify sequenced SARS-CoV-2 genomes with sequencing errors. They explore several feature encodings and machine learning methods to identify such erroneous sequences.

**Summary Of The Review:**

While the paper has a couple of good ideas, it is hard to follow, does not introduce and validate novel concepts and methods, and models the errors in an unrealistic manner.

---

> ### Author Response · Authors · 2021-11-21
> **Response to Reviewer YVK3**
>
> (a) Comment: My major issue with this paper is the sequencing errors are not modeled realistically. Most Illumina sequencers have error rates of less than 0.5% of nucleotides [PMID:33817639], the authors introduce at least 5% error rates in the amino acids (each amino acid is encoded by three nucleotides). NB. Interestingly, this is the citation that the authors use to justify their “simulated” error rates. The authors mention PacBio sequencing errors, but I see no evidence of that in the text.
>
> •Response: Thank you for pointing this out. This is true that we are not using any existing technology such as Illumina or PacBio to introduce error in the sequences. Initially, our goal was to see the behavior of classifiers on random error. In future, we will include results for errors computed using these sequences techniques.
>
> (b) Comment: Although spike protein is 1,273 amino acids long; however, there are around200 amino acids in the receptor-binding domain (RBD) of the spike protein that is of special interest [PMID: 34075212]. A more realistic analysis would focus mostly on the RBD. The community is increasingly generating more high-throughput datasets assaying different aspects of the RBD and ACE2 binding, which can help researchers train more accurate models,e.g., [PMID: 32841599].
>
> •Response: This is an interesting point which we will keep in mind. This might even be useful for having a more compact feature vector representation in other studies.
>
> (c) Comment: The concepts introduced in the paper are not novel and have been studied in detail: encoding of amino acids, k-mer-based models, and simple MLPs have been thoroughly studied in computational genomics.
>
> •Response: The goal of this study is to analyze the limitations of the existing models. This is true that encoding of amino acids has been studied before, but our goal is to benchmark the performance of ML and DL algorithms on those existing encoding meth-ods. We find out that introducing errors in the sequences does not effect (significantly) the performance of the DL algorithm. Keeping in mind that there are only a few mutations for each coronavirus variant, this robustness testing shows that DL algorithm is still able to differentiate between different variants.
>
> (d)Comment: The text is hard to follow and has quite a few grammatical errors/missing words starting from the abstract. A thorough review and polish of the text would improve the paper.
>
> •Response: We thank the reviewer for highlighting this. We will improve the presentation of the manuscript in the future.

---

### Decision · Program_Chairs · 2022-01-20

**Decision:**

Reject

**Comment:**

This paper presents a framework to test the accuracy and robustness of different machine learning algorithms in classifying the COVID-19 spike sequences. After reading the paper and taking into consideration the reviewing process, here are my comments:
- The work is aligned to the efforts on understanding the COVID-19 pandemic.
- Many concepts are not novel.
- Sequences errors are not modeled in a realistic way.
- The benchmark is very limited and nonlinear machine learning approaches are presented.
- Many typos are presented.
From the above, the paper is not suitable for aacceptance in ICLR.